



# Geochemical mass-balance, weathering and evolution of soils formed on a Quaternary-age basaltic toposequences

Hüseyin ŞENOL[1*], Tülay TUNÇAY[2], Orhan DENGİZ[3]

[1]Suleyman Demirel University, Faculty of Agriculture, Department of Soil Science and Plant Nutrition, Isparta, Turkey.
[2]Soil Fertilizer and Water Resources Central Research Institute, Ankara, Turkey.
[3]Ondokuz Mayıs University, Faculty of Agriculture, Department of Soil Science and Plant Nutrition, Samsun, Turkey.

*Correspondence to*: Huseyin Senol (*huseyinsenol@sdu.edu.tr*)

**Abstract.** The purpose of this research is to assess the geochemical mass-balance and weathering intensity of Typic
Haplustert and Lithic Ustorthent soils represented by four profiles that developed in a Quaternary-age basaltic toposequence under semi-humid conditions in the central Black Sea region of Turkey. The researchers employed mass-balance analysis with a view to measuring elemental gains and losses along with alterations concerning the soils formed on the basaltic parent material. For this end, geochemical properties, elemental mass-balance changes and certain physicochemical features were identified to benchmark the weathering levels of the profiles. Lithic Ustorthents are distinguished by having a rough texture
along with a low organic substance ingredient, whereas Typic Haplusterts have a high clay texture with low bulk density and slickenside features. X-ray diffraction showed that smectites were the prevailing minerals inside the Typic Haplusterts, while a significant amount of kaolinite and illite was observed in the Lithic Ustorthents. Mass-balance computations indicated that massive mineral weathering resulted in substantial Si losses through leaching as well as an exchange of cations, such as $Na^+$, $K^+$, $Ca^{2+}$ and $Mg^{2+}$, particularly from the upper horizons. The study also took into account other features such as the
pedogenic evolution of soils using weathering indices such as CIA, CIW, bases/R2O3, WIP, P and PIA. According to the results, CIA, CIW, PIA, P, WIP and bases/R2O3 index values of all soils varied between 42.33 to 73.83, 44.46 to 80.43, 37.53 to 65.63, 75.39 to 84.31 and 0.45 to 1.27 respectively, to solum depth. This result indicated that soils classified as Entisol and Vertisol have similar pedochemical properties. In spite of similar weathering rates, the soils were classified under different groups as a result of erosion. This showed that the conditions for soil development in the studied area had a
far more impact on weathering and elemental loss than the parent material on the site.

**Key Words:** soil formation, mass-balance, toposequences, weathering indices, clay minerals

## 1. Introductıon




Soil is viewed as an open system with additions and removals of materials to and from the profile, and translocation and transformation within the profile. Pedogenesis can be progressive or regressive. Progressive pedogenesis includes processes that promote differentiated profiles leading to horizontal separation, leaching, developmental upbuilding and soil deepening. In this area particularly, soil toposequences are frequently utilized to indicate the relative soil development degree within

changing durations of soil formation, taking the fact that the other soil formation factors resemble each other into account. During weathering, the elements are moved and re-delivered in different ways as there are many pedogenic processes that influence different elements and produce different results. These processes include dissolution of primary minerals, formation of secondary minerals, redox processes, transport of material and ion exchange (Middleburg *et al.*, 1988, Dengiz *et al.*, 2013). It is therefore important to understand how a soil is formed from bedrock and to examine how chemical or

physical weathering influences the geochemical evolution of soils. According to Birkeland (1999, it is both possible to re-deliver down-slope elements by weathering as a function of their mobility under steady ongoing changes in the geochemical environments along the slope. Soil development, in the chemical sense, is also roughly synonymous with weathering (Bohn *et al.*, 1985). Therefore, weathering rates, or element depletion rates or accumulation in the catchment area or soil profile scale, are often acquired by mass-balance studies (input–output budgets) and provide important indications concerning the

reactivity of a relevant soil or catchment area (Dahms *et al.*, 2012). Moreover, regressive pedogenesis (Minasny *et al.*, 2008; Sommer *et al.*, 2008) includes steps that promote rejuvenation processes, retardant upbuilding (soil impedance produced by surface-accreted materials) and surface removals (erosion). In this case, the slope position has a key role in soil formation, and the significance of the effects of the slope gradient on the soil's physicochemical and mineralogical properties has been widely reported (Jong *et al.*, 2000, Dengiz, 2007). Soil formation is therefore considered as a net change in mass-balance of

the soil compartment.

The researcher developed certain mass-balance equations taking these properties into account. These equations aimed to determine soil deformation (strain), and open chemical system gains and losses in the soil, in relation to the parent material. It is also possible to conduct a soil mineralogy examination to determine whether it is pedogenesis or differences in parent materials that lead to chemical differences. Volumetric and mass changes during soil formation were assessed by

administering a mass conservation equation (Brimhall and Dietrich, 1987; Brimhall *et al.*, 1988, 1991; Chadwick *et al.*, 1990).

This study mainly aimed to describe an empirical method with a view to modeling this open system contribution in accordance with the compositional change analysis within soil profiles. This is done by determining the weathering rates and mass-balance of soils developed on a catena with respect to the soil-forming factor topography, and using some weathering

indices based on geochemical information in semi-humid environments with similar geological patterns and climate. The mass-balance, and certain weathering indices with other features – such as the mineralogy and some of the analytical characteristics of soils formed over a toposequence and Quaternary-age basaltic parent materials – are indicated here to discuss their use in quantifying the phases of maturity and the duration of soil formation.




## 2. Materials and Methods

### 2.1. Site Description

This study was carried out along a transverse section between the Bafra Plain and the Canik mountain, and located approximately 20 km west of the Samsun province in the central Black Sea Region of Turkey (Figure 1). The study area is situated between the coordinates 4597065 N –253437 E and 4595005 N – 251693 E (UTM/WGS 84 m).

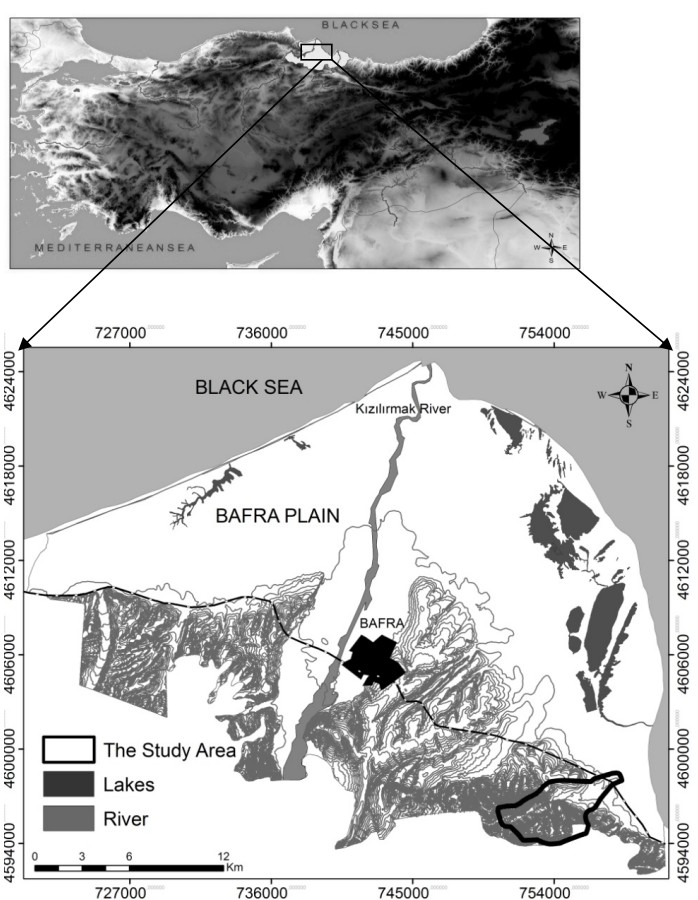

Figure 1. Location map of the study area





The study area ranges from 10 m to 300 m and includes four landscape features (foot slope, back slope, lowland plateau and shoulder) representing changes in geomorphology, topographic gradient, parent material and soil characteristics. The underlying bedrock consists primarily of Quaternary-age basaltic, colluvial deposits on the foot slope, lowland plateau, and

Mesozoic-age basalt and marl-limestone on the back slope and highland plateau. Today, the region has a semi-humid climate. Summers are warmer than winters (Avg. Temperatures: July, 22.2 °C; January, 6.9 °C). The average annual temperature, rainfall and evaporation are 13.6 °C, 764.3 mm and 726.7 mm, respectively. Soil temperature and water moisture regimes at the study site were classified in accordance with the Soil Survey Staff (Soil Taxonomy, 1999) as mesic and ustic, respectively. Physiographically, the area comprises four main units. The study area is covered predominantly by

pasture and forest land. A minority of the site consists of a slightly sloped (0.0 to 2.0%) low plateau, while other sections are hilly and moderately to severely sloped (3 to 20%). Only a small part of the foot slope and lowland plateau is agricultural land.

**2.2. Soil Sampling and Analysis**

Based on the hypothesis that topography, parent material and climate–vegetation cover might be the main controlling factors

for mass-balance in soil development, soils have been studied along a transverse section (diagonally in the southwestern to northwestern direction) using four representative profiles (Figure 2). The morphological features of these four profiles from the field were identified and sampled by genetic horizons and classified in accordance with Soil Survey Staff (1993, 1999). Twelve disturbed and undisturbed the samples of soil were taken to the laboratory to search for their physical, chemical and mineralogical features. The soil samples were first air-dried and then passed through a 2 mm sieve to be ready for laboratory

analysis.

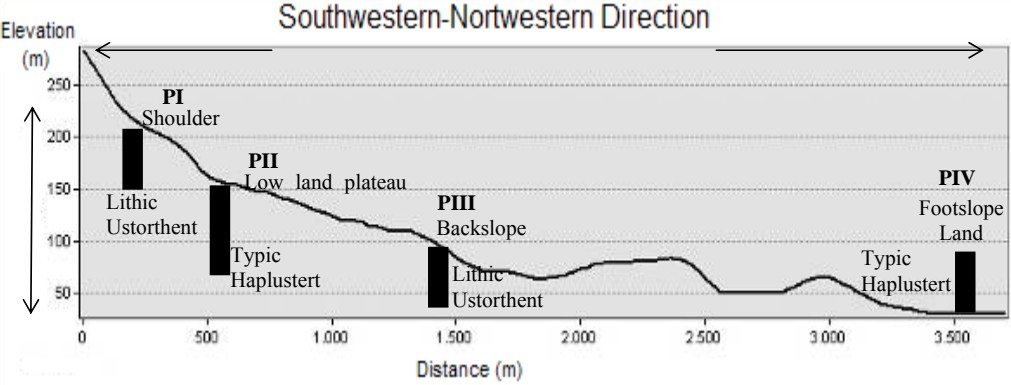

Figure 2.  Transect of the four different soil profiles on the same parent material but different topographic positions.




### 2.3. Physical, Chemical and Mineralogical Analysis

After the soil samples were air-dried and passed through a 2 mm sieve, their particle size distribution was designated by the
hydrometer method (Bouyoucos, 1951), following the removal of organic matter with 30% $H_2O_2$, the removal of sulfates by
leaching the salts with distilled water, the removal of carbonates with 1 M NaOAC at pH 5, and sample dispersion by
agitating it in 10 ml of 40% sodium hexametaphosphate (Calgon) (Gee and Bauder, 1986). Bulk density (Blacke and Hartge,
1986) was determined from the undisturbed samples. Organic matter and total nitrogen content were determined in the air-
dried samples employing the Walkley-Black wet digestion method (Nelson and Sommers, 1982). The pH and EC-electrical
conductivity (of the saturated sample) were determined by using a Soil Survey Laboratory method (2004). The lime content
was determined by a Scheibler calcimeter (Soil Survey Staff, 1993). Exchangeable cations and cation exchange capacities
(CEC) were measured utilizing a 1 N $NH_4OAC$ (pH 7) method (Soil Survey Laboratory, 2004).

The clay fraction ($< 2\,\mu m$) was acquired from the soil, after destruction of organic matter with dilute and Na-acetate-buffered
$H_2O_2$ (pH 5), by dispersion with Calgon and sedimentation in water. Prepared specimens on glass slides were analyzed by X-
ray diffraction using Cu Kα radiation from 2° to 30° 2θ with steps of 0.02° 2θ at 2 s step$^{-1}$. The following treatments were
carried out: Mg saturation, ethylene glycol solvation (EG) and K saturation, followed by heating for 2 h at 550 °C. Minerals
and their relative abundance were determined by their diagnostic XRD spacing and assessed using their XRD relative peak
intensities from the XRD scheme (Whittig and Allardice, 1986).

### 2.4. Mass-Balance

Mass-balance models provide a means to account for the fate of elements during soil evolution by measuring the gains,
losses and transformations that emerge over time (Brimhall and Dietrich, 1987). Long-term weathering rates of soils were
obtained from the calculations of the enrichment/depletion factors determined by utilizing immobile element content such as
Ti, Zr or V (Egli et al., 2008). In this study, Ti was used as an immobile element. Volumetric changes that emerge during
pedogenesis were determined by adopting the classical definition of strain, εi, w (C) (Brimhall and Dietrich, 1987).

$$\varepsilon_{i,w} = \frac{\Delta z_w}{\Delta z} - 1$$

(1)

where $\Delta z_w$ is the weathered equivalent height (m), $\Delta z$ is the columnar height (m), a positive value indicating enrichment and
a negative value indicating depletion. The calculation of the open-system mass transport function (T), (Chadwick et al.,
1990) is shown below:





$$\tau_{j,w} = \left( \frac{\rho_w c_{j,w}}{\rho_p c_{j,p}} \left( \varepsilon_{i,w+1} \right) \right) - 1 \qquad (2)$$

$\tau_{j,w}$ is the calculation of the open-system mass transport function; $\rho_w$ is the bulk density of the weathered soil, $C_{j,w}$ is the concentration of element in the weathered product, $C_{j,p}$ is the concentration of the element in the protolith (e.g. the unweathered parent material or bedrock) $(kg.t^{-1})$ and $\rho_p$ is the bulk density of the protolith $(t.\,m^{-3})$ (Egli and Fitze, 2000).

$$m_{j,flux(z_w)} = \sum_{a=1}^{n} C_{j,p} \rho_p \left( \frac{1}{\varepsilon_{i,w} + 1} \right) \tau_{j,w} \Delta z_w \quad (3)$$

$m_{j,flux(zw)}$, with n soil layers is the calculation of changes in the mass of element j and is given by (Egli and Fitze, 2000) $(g.cm^{-2})$. Positive values indicate additions to the system, whereas negative values indicate losses from the system.

## 3. Results and Discussion

### 3.1 Soil morphological properties and soil classification

Some morphological properties of the soil and soil classifications for the soil profiles are shown in Table 1. The four different soil profiles have the same parent material, but at different topographic positions on the transverse section of the study area and showed changes in terms of their depth, color, structure and textural distribution. Many researchers indicated

that different morphological, physical and chemical features of soils can be related to various topographic positions or specific landforms associated with accumulation, erosion and runoff processes during soil formation, due to the effects of adding water and energy, and taking them away from, the soil (Birkeland, 1999; Yair, 1990; Dahlgren *et al.*, 1997; Canton, 2003; Dengiz, 2010). Soil texture is a major property of soil, and it may be influenced by geological processes such as erosion. The pedological differences along the transverse section are also shown in Figure 2. Clay percentages of the profiles

affected by slope gradient showed variations from 32% to 68% with increasing elevation This result, to some extent, concurred with the result of Kreznor *et al.*, (1989), Rezaei and Gilkes (2005) Dengiz *et al.*, (2013). It was found that particle size depended on landscape attributes, including slope degree or gradient. This change also has an effect on the structural development of soils along the transverse section. The soils formed on the lowland plateau and foot slope have a clay texture, while the soils formed on a high degree of slope have a clay loam texture. These differences represent the obvious

effects of erosion, whereby surface soils have been carried from the back slope to the foot slope and lowland plateau, their accumulation leading to progressively darker, deeper and finer-textured soils with decreases in elevation. The clay fraction, particularly, tended to increase with depth in all profiles up to the Bss horizon in profiles PII and PIV. This case also encourages developing soil structure, and Bss horizons were characterized by a strong blocky and prismatic structure with slickenside formation. As for the PI and PIII profiles, the slope degree played an significant role in supervising the soil





formation and development. Due to movement of fine earth and organic matter, structural growth of the back slope and shoulder position soils on the surface horizon is weak and also moderate, fine and granular (1fgr-2mgr).

The Soil Taxonomy (Soil Survey Staff, 1999) and the FAO/UNESCO soil map of the world legend (FAO/ISRIC, 2006) was used to classify the soils investigated. The classification was based on the results of the morphological, physical and chemical characteristics of the soils. The soils located on the foot slope and lowland plateau were formed from a basalt parent material and contained more than 40% clay, as indicated by surface cracks ranging from 1 to 5 cm in width, as well as intersecting slickensides and shiny pressure faces in the subsurface horizon (Bss) This also reflects a shrinking and swelling of the soil. Accordingly, profiles PII and PIV were classified as Typic Haplustert. These profiles were also classified as Haplic Vertisol by taking the FAO/ISRIC (2006) classification system into consideration.

Table 1. Morphological properties and classification (Soil Taxonomy, 1999 and FAO/ISRIC, 2006) of profiles

| Horizon | Depth (cm) | Color (dry) | Color (moisture) | Structure | Boundary | Special features |
|---|---|---|---|---|---|---|
| PI / Shoulder | *(Lithic Ustorthent / Eutric Regosol )* | | | | | |
| A | 0-16 | 2.5 Y 5/2 | 2.5 Y 3/2 | 2fgr | cs | - |
| Cr | 16+ | 2.5 Y 3/3 | 2.5 Y 5/2 | sg | - | - |
| PII / Low land plateau | *(Typic Haplustert / Haplic Vertisol)* | | | | | |
| A | 0-12 | 10 YR 5/3 | 10 YR 3/4 | 3mgr | cs | cracks |
| Bss1 | 12-48 | 10 YR 5/3 | 10 YR 4/3 | 2msbk | cw | slickenside |
| Bss2 | 48-89 | 10 YR 5/3 | 10 YR 3/2 | 3mpr | cw | slickenside |
| C | 89+ | 10 YR 4/3 | 10 YR 3/2 | mas | - | - |
| PIII / Backslope | *(Lithic Ustorthent / Eutric Regosol)* | | | | | |
| Ap | 0-24 | 2.5 Y 5/3 | 10 YR 3/2 | 2mgr | as | - |
| Cr | 24+ | 2.5 Y 4/3 | 2.5 Y 4/3 | sg | - | - |
| PIV / Footslope | *(Typic Haplustert / Haplic Vertisol)* | | | | | |
| Ap | 0-23 | 2.5 Y 5/2 | 2.5 Y 3/3 | 3mgr | as | cracks |
| Bss1 | 23-65 | 2.5 Y 5/3 | 2.5 Y 3/2 | 3mpr | cw | slickenside |
| Bss2 | 65-106 | 2.5 Y 5/2 | 2.5 Y 4/1 | 3cpr | cw | slickenside |
| C | 106+ | 10 YR 6/1 | 10 YR 4/2 | mas | - | - |

**Abbreviations:** Boundary: a = abrupt; c = clear; g = gradual; d = diffuse; s = smooth; w = wavy; i = irregular
Structure: 1 = weak; 2 = moderate; 3= strong; sg = single grain; mas = massive; vf = very fine; f = fine; m =medium; c = coarse;
gr = granular; pr = prismatic; abk = angular blocky; sbk = subangular blocky

Profiles PI and PIII were classified as Lithic Ustorthent. The slope was one of the most significant factors controlling the pedogenic process in this pedon, which is located on a high back slope. Slope brings greater runoff, as well as to greater



translocation of surface materials down-slope through surface erosion and soil movement. The horizon orders of the PI and PIII profiles were defined as A-Cr horizons. This means that these profiles had no diagnostic subsurface horizons and included a lithic layer within 50 cm of soil depth. Therefore, this soil is classified as a young soil due to low pedogenetic development. According to the FAO/ISRIC (2006) classification system, these profiles were classified as Eutric Regosol

### 3.2. Physical and chemical soil properties

Physical and chemical properties for the soil profiles are presented in Table 2. Clay content ranged from 32.1% to 68.5%, while sand content varied from 15.8% to 40.1% in the profiles. The textural class of the soils changed between clay and clay loams. Typic Haplustert (PII and PIV profiles) included higher fine fraction content than those of silt and sand. Because these profiles were located on low slope positions (the lowland plateau and foot slope) in the study field, they were within the accumulation area. On the other hand, other profiles formed on high slope positions that led to soil transport. The reason for it is higher clay deposition in these profiles through the processes of transportation and colluviation, rather than weathering from the adjacent hill areas (Akhtaruzzaman *et al.*, 2014). The clay content was lower in the A horizons and increased with depth in all profiles. In addition to that, Ovalles and Collins (1986) also reported that physical soil properties such as the distribution of clay content with depth, sand content and pH were highly correlated to landscape position.

Bulk density values for the soils ranged from 1.22 to 1.61 $gr.cm^{-3}$ across the profiles of the toposequence. Bulk density content was relatively lower in different horizons of profile PIV compared to that of other profiles and included high amounts of sand particles. The low amount of organic carbon (Gupta et al., 2010) might have contributed to the higher values of bulk density in the soil. The bulk density values were the lowest in the surface or close to the surface horizons, and they increased progressively with depth. Similar results were reported by Göl and Dengiz (2008). Lower bulk density values were related to organic matter and clay content, especially at the surface horizon.

Organic matter originates from open-system influxes, which decreased with depth. The amount of organic matter added to surface horizons varied between 1.65% and 2.35%, also influx decreased from 0.14% to 0.15% in the C/Cr horizons. Soil pH does not vary significantly along the toposequence. The pH valued ranged from 6.93 to 8.25 in all horizons and increased with depth, except for Lithic Ustorthent soils (PI). This profile has lower pH values, being located on the upper slope positions that promote basic cation leaching, whereas other profiles were neutral to basic soil reaction pH values. In addition, $CaCO_3$ contents ranged from 0.20% to 2.67% even the development of all these profiles on basaltic parent material which doesn't produce carbonate. It can be said that this case resulted from carbonate contamination. Similar to our findings, Aksoy (1991) found a lime content of 2% to 26% in soils on the Kayacik plains in Gaziantep, although the primary material in the area of their study was basalt. The CEC values varied and could be correlated to clay and organic matter content, with maximum values of 42.80 $cmolc.kg^{-1}$ in the surface soils of all profiles, and minimum values of 11.09 $cmolc.kg^{-1}$ in the subsurface horizons (C or Cr); additionally they showed no trend with depth except for the PIV profile that showed no significant decrease in value with depth. The highest CEC values were found where smectite was the predominant clay type in the presence of aluminosilicate minerals. The base saturation values were almost 100% for all profiles. Calcium and



Magnesium were the prevailing exchangeable cations for all profiles and ranged from 33.16 to 42.24 cmolc. kg$^{-1}$ in surface horizons. Higher contents of Ca and Mg in the surface horizons were probably related to weathering of the basalt parent material and biological accumulation from plants (Akbar *et al.*, 2010).

Table 2. Some physical and chemical features of studied profiles

| Horizon | Depth (cm) | pH(H2O) (1/2.5) | EC (dS.cm$^{-1}$) | CaCO$_3$ (%) | O.M (%) | Exchangeable Cations (cmolc·kg$^{-1}$) | | | CEC (cmolc·kg$^{-1}$) | B.D (gr.cm$^{-3}$) | Particle size distribution (%) | | | |
|---|---|---|---|---|---|---|---|---|---|---|---|---|---|---|
| | | | | | | Na | K | Ca+Mg | | | C | Si | S | Class |
| PI / Shoulder | | | *(Lithic Ustorthent / Eutric Regosol )* | | | | | | | | | | | |
| A | 0-16 | 7.03 | 0.19 | 0.50 | 2.25 | 0.28 | 1.02 | 33.16 | 34.46 | 1.44 | 34.4 | 25.5 | 40.1 | CL |
| Cr | 16+ | 6.93 | 0.25 | 0.20 | 0.42 | 0.45 | 1.24 | 9.39 | 11.09 | 1.61 | - | - | - | - |
| PII / Low land plateau | | | *(Typic Haplustert / Haplic Vertisol)* | | | | | | | | | | | |
| A | 0-12 | 7.05 | 0.16 | 0.79 | 1.71 | 0.35 | 0.24 | 40.17 | 40.76 | 1.42 | 41.5 | 24.2 | 34.3 | C |
| Bss1 | 12-48 | 7.72 | 0.19 | 0.29 | 1.69 | 0.74 | 0.31 | 48.07 | 49.13 | 1.22 | 68.5 | 18.3 | 23.9 | C |
| Bss2 | 48-89 | 7.79 | 0.34 | 1.37 | 0.59 | 1.31 | 0.41 | 47.25 | 48.97 | 1.34 | 49.8 | 26.4 | 20.7 | C |
| Cr | 89+ | 7.96 | 0.30 | 1.18 | 0.17 | 1.26 | 0.24 | 32.84 | 34.35 | 1.44 | - | - | - | - |
| PIII / Backslope | | | *(Lithic Ustorthent / Eutric Regosol)* | | | | | | | | | | | |
| Ap | 0-24 | 7.87 | 0.55 | 0.49 | 2.35 | 0.41 | 0.28 | 42.24 | 42.93 | 1.44 | 32.1 | 27.9 | 40.1 | CL |
| Cr | 24+ | 8.04 | 0.10 | 0.29 | 0.55 | 1.03 | 0.15 | 14.20 | 15.38 | 1.65 | - | - | - | - |
| PIV / Footslope | | | (Typic Haplustert / Haplic Verisol) | | | | | | | | | | | |
| Ap | 0-23 | 7.50 | 0.17 | 0.20 | 1.65 | 0.22 | 1.67 | 40.91 | 42.80 | 1.28 | 56.2 | 23.1 | 20.7 | C |
| Bss1 | 23-65 | 7.30 | 0.44 | 0.98 | 1.26 | 0.25 | 1.47 | 39.64 | 41.36 | 1.29 | 62.6 | 12.8 | 24.5 | C |
| Bss2 | 65-106 | 8.25 | 0.17 | 1.10 | 1.09 | 1.33 | 1.41 | 37.59 | 40.33 | 1.23 | 68.4 | 15.8 | 15.8 | C |
| C | 106+ | 8.14 | 0.11 | 2.67 | 0.14 | 1.35 | 1.40 | 36.04 | 39.78 | 1.30 | - | - | - | - |

EC: Electrical Conductivity, BD: Bulk Density, O.M: organic Matter, CEC: Cation Exchange Capacity

### 3.3. Total content

The composition of the material that was investigated clearly reflects an Si-rich and basaltic character. The major element concentrations and bulk density of the toposequences profiles developed on Quaternary and Mesozoic basalt (colluvial deposit; foot slope, back slope, lowland plateau and slope, respectively) in the Samsun-Bafra Plain, Northern Turkey, are given in Table 3. All soils contained large concentrations of SiO$_2$, Al$_2$O$_3$ or Fe$_2$O$_3$. These elements Si, Al, Fe were chosen

15 due to their abundant existence in the soils (>93% of total elemental composition occurs as oxides) (Langley-Turnbaugh and Bockheim, 1998).



In most cases, chemical compositions of parent materials of soils resembled each other. There were, however, some minor differences in the chemical composition of the B and C horizons between the sites. The $SiO_2$ had a slightly higher content at the PIV site; the $Al_2O_3$ had a slightly lower content in the PI and PIII profiles; the $Fe_2O_3$ had a slightly higher content in the PIII soil profiles; the MgO had a slightly lower content in the soil profiles of PIV and PII, and the CaO had a slightly higher

content in the soils of PIV – especially in Bss 2 and C horizons – and in PI.

The content of $Fe_2O_3$ in all soil profiles was determined to be at a higher level than their parent materials. On the other hand, the content of other elements in the soils were detected at a lower level than those of the parent materials (except for the PII profile for $SiO_2$). In the solum layer of Typic Haplustert (PII), concentrations of Al, Fe and Mg are enriched, whereas concentrations of Si, Ca, Na and K levels were decreased (except for the PIV profile where Si and K increased in the solum

layer).

Table 3. Geochemical characteristics (total analysis of the bulk material including soil skeleton and fine earth) of the investigated profiles

| Horizon | Depth (cm) | Bulk Density | SiO2 (%) | Al2O3 (%) | Fe2O3 (%) | MgO (%) | CaO (%) | Na2O (%) | K2O (%) | TiO2 (%) | P2O5 (%) | MnO (%) | Cr2O3 (%) |
|---|---|---|---|---|---|---|---|---|---|---|---|---|---|
| PI / Shoulder | *(Lithic Ustorthent / Eutric Regosol )* | | | | | | | | | | | | |
| A | 0-16 | 1,44 | 45,32 | 6,99 | 11,83 | 4,16 | 5,06 | 0,84 | 2,85 | 0,84 | 0,48 | 0,22 | 0,012 |
| Cr | 16+ | 1,61 | 45,80 | 7,46 | 10,02 | 4,85 | 5,90 | 0,92 | 2,80 | 0,78 | 0,47 | 0,25 | 0,008 |
| PII / Low land plateau | *(Typic Haplustert / Haplic Vertisol)* | | | | | | | | | | | | |
| A | 0-12 | 1,42 | 50,73 | 17,56 | 10,69 | 2,34 | 1,42 | 1,21 | 2,93 | 0,98 | 0,17 | 0,22 | 0,021 |
| Bss1 | 12-48 | 1,22 | 46,30 | 17,36 | 11,02 | 5,44 | 1,48 | 1,38 | 3,08 | 0,82 | 0,45 | 0,18 | 0,009 |
| Bss2 | 48-89 | 1,34 | 53,67 | 17,78 | 8,63 | 2,25 | 1,60 | 1,14 | 2,56 | 0,87 | 0,08 | 0,18 | 0,024 |
| C | 89+ | 1,44 | 58,32 | 16,13 | 7,81 | 1,74 | 1,70 | 1,59 | 3,14 | 1,06 | 0,18 | 0,19 | 0,035 |
| PIII / Backslope | *(Lithic Ustorthent / Eutric Regosol)* | | | | | | | | | | | | |
| Ap | 0-24 | 1,44 | 43,83 | 10,91 | 14,04 | 3,89 | 3,19 | 0,37 | 1,75 | 1,02 | 0,33 | 0,50 | 0,006 |
| Cr | 24+ | 1,65 | 42,83 | 9,75 | 13,92 | 3,71 | 5,72 | 0,55 | 1,29 | 1,07 | 0,40 | 0,57 | 0,002 |
| PIV / Footslope | (Typic Haplustert / Haplic Verisol) | | | | | | | | | | | | |
| Ap | 0-23 | 1,28 | 59,38 | 13,76 | 7,02 | 1,53 | 2,37 | 1,25 | 1,62 | 0,97 | 0,20 | 0,21 | 0,060 |
| Bss1 | 23-65 | 1,29 | 58,87 | 13,97 | 7,10 | 1,60 | 1,05 | 1,05 | 1,43 | 0,98 | 0,06 | 0,29 | 0,052 |
| Bss2 | 65-106 | 1,23 | 53,20 | 13,11 | 6,48 | 2,30 | 8,22 | 0,99 | 1,37 | 0,87 | 0,10 | 0,17 | 0,045 |
| C | 106 + | 1,30 | 52,78 | 10,05 | 6,39 | 2,51 | 8,55 | 1,04 | 1,42 | 0,84 | 0,12 | 0,14 | 0,043 |

The chemical composition for all soil samples was predominantly $SiO_2$ and $Al_2O_3$, with $SiO_2$ from 42.83 to 59.38% and $Al_2O_3$ from 6.99 to17.78%. $TiO_2$ content, 0.78 to 1.07%, was markedly decreased in the weathered products. For this case,



Jianwu et al., (2014) also reported that fresh basalt had a decreased titanium content of between 0.82 to 0.97% depending on the parent material weathering. The soil samples showed significant variations of 6.99 to 17.78% in terms of the $Al_2O_3$ concentration as a result of the abundance of plagioclase. This result was also supported by Machado et al., (2008). Major element concentration of the PI and PII profiles showed a trend of irregular distribution with depth. However, other profiles
tended to have a regularly decreasing distribution with depth in their major element concentrations. The CaO values were higher in the subsoil than the surface. The MgO values showed that there were significant differences among the horizons. The high value of MgO was caused by the presence of mica minerals (biotite). In addition, concentrations of $K_2O$, 1.29 to 3.08%, and $Na_2O$, 0.84 to 1.38%, correlated to the existence of alkali feldspar and common mineral in the basaltic rock.

**3.4. Chemical weathering indices**

Based on the principles of soil genesis, alkali and alkaline earth metals (Na, Mg, K and Ca) move through the soil horizons before the silicium fraction as the weathering process continues (Souri *et al*., 2006). Although profiles developed on basaltic materials are still new (Quaternary age), the evaluation of weathering indices was performed with the objective of showing basaltic material at an early stage of weathering. Weathering indices for weathering profile samples have conventionally
been measured using the molecular proportions of major element oxides (Price and Velbel, 2003). Potential chemical weathering indices of the unweathered (C/Cr horizon) and weathered (solum) basaltic material are shown in Table 4.

Nesbitt and Wilson (1992) separated weathering profiles into four qualitative analyses, which are incipient, intermediate, advanced and extreme. Several indices are considered, such as the chemical index of alteration (CIA; Nesbitt and Young, 1982), the Chemical Index of Weathering (CIW; Harnois, 1988), the Weathering Index of Parker (WIP), the Plagioclase
index of alteration (PIA; Fedo *et al*., 1995), the Product index (P; Reiche, 1950), the bases/$R_2O_3$ (Birkeland,1999) and the Vogt index/Vogt's residual index (V; Vogt, 1927). They can quantify the condition of different parent materials at the onset of weathering, assess their fertility, provide a better perception of elemental mobility in the course of weathering and estimate the source of soil nutrients in addition to changes in the primary minerals (Fiantis *et al*., 2010).

These indices should be easy to use, involving chemical elements common in soil analyses (Harnois, 1988; Price and Velbel,
2003). The use of the CIA, CIW and PIA are to give a quantitative measure of feldspar weathering by relating Al enrichment in the weathering residues – in contrast to Na, Ca and K – which should be removed from a soil profile during plagioclase or K-feldspar weathering (Nesbitt and Young, 1982; Price and Velbel, 2003; Fiantis *et al*., 2010). WIP, introduced by Parker (1970) for silicate rocks is based on the proportions of alkaline and alkaline earth metals present (Fiantis *et al*., 2010). The most mobile major elements don't need to suppose that sesquioxide concentration remains approximately constant in the
course of weathering. The V index is a geochemical method for evaluating the maturity of residual sediments. The Bases/$R_2O_3$ index uses the principle that, in a leaching process, the ratio between the concentration of mobile base elements (Mg, Ca, Na and K) and relatively immobile elements (Al, Fe and Ti) decreases over time (Mourier, 2008).

Table 4. Weathering rates of studied soils





| Horizon | Depth (cm) | CIA | CIW | WIP | PIA | P | Base/R2O3 | V |
|---|---|---|---|---|---|---|---|---|
| PI / Shoulder | *(Lithic Ustorthent / Eutric Regosol )* | | | | | | | |
| A | 0-16 | 55.73 | 62.02 | 55.39 | 45.58 | 76.33 | 0.96 | 0.96 |
| Cr | 16+ | 53.64 | 59.17 | 59.65 | 44.31 | 75.30 | 1.08 | 0.84 |
| PII / Low land plateau | *(Typic Haplustert / Haplic Vertisol* | | | | | | | |
| A | 0-12 | 65.37 | 74.15 | 47.21 | 53.54 | 77.32 | 0.60 | 1.72 |
| Bss1 | 12-48 | 60.38 | 65.79 | 48.81 | 50.68 | 75.39 | 1.21 | 1.75 |
| Bss2 | 48-89 | 60.87 | 79.70 | 41.17 | 59.80 | 78.72 | 0.53 | 2.00 |
| C | 89+ | 64.82 | 75.10 | 48.51 | 51.12 | 71.30 | 0.58 | 1.80 |
| PIII / Backslope | *(Lithic Ustorthent / Eutric Regosol)* | | | | | | | |
| Ap | 0-24 | 58.51 | 63.87 | 36.77 | 61.24 | 72.79 | 0.65 | 1.22 |
| Cr | 24+ | 59.81 | 62.61 | 40.24 | 55.35 | 71.29 | 0.75 | 0.98 |
| PIV / Footslope | (Typic Haplustert / Haplic Verisol) | | | | | | | |
| Ap | 0-23 | 61.93 | 67.70 | 34.00 | 53.39 | 84.31 | 0.63 | 1.45 |
| Bss1 | 23-65 | 73.83 | 80.43 | 27.63 | 65.63 | 83.30 | 0.45 | 2.08 |
| Bss2 | 65-106 | 42.33 | 44.46 | 46.87 | 37.53 | 82.92 | 1.27 | 0.66 |
| C | 106+ | 41.25 | 43.36 | 49.12 | 36.38 | 82.93 | 1.35 | 0.62 |

Generally, the weathering indices of both the unweathered and weathered basaltic materials are quite similar; but, the unweathered parent material has slightly lower values than weathered solum values for CIA, CIW, PIA, P and V, however, it has higher values for WIP and bases/$R_2O_3$.

5   The CIA value of unweathered basalt is 41.25%, while those of soil samples range from 42.33% to 73.83%, and no significant differences were found between the profiles, except for profile PIV. The highest CIA value was determined in the Bss1 horizon of profile PIV, whereas the lowest value was found in the Bss2 horizon of Typic Haplustert.

Similarly, CIW values have been shown to correlate to the CIA value. CIW value of unweathered basalt is 43.36%, while those of soil samples vary from 44.46% to 80.43% and there were no significant differences between profiles, except for

10  profile PIV. Low and high CIW values were seen in the same profile, matching the CIA values.

In contrast, the WIP values for the same profile samples yielded the opposite trends, showing decreasing values in the PIV profile. The highest WIP value was in the Cr horizon of Lithic Ustorthent (PI), and the lowest value was, again, in the Bss1 horizon of profile PIV. The WIP indices ranged between 27.63 and 59.65. The PI and PIII profiles have the weakest silicate weathering, while the PII and PIV profiles have the strongest weathering. Typic Haplustert (PII and PIV profiles) primarily





undergo weak and moderate silicate weathering, respectively. On the other hand, the degree of silicate weathering in the Lithic Ustorthent soils was in the intermediate stage.

Based upon the principles of the Parker (PIA) and Product (P) indices, the samples are more weathered and regarded to be older. These indices have quite different chemical bases. While Parker (PIA) is a suitable index to seek for the mobility of

alkali and alkaline earth metals through the soil profile, the Product index is appropriate to track movements of less mobile elements (Souri *et al*., 2016).

In the profiles, the PIA value varied between 23.38 and 65.63, and no significant differences were found between the profiles. The highest PIA rate was in the Bss1 horizon of profile PIV, and the lowest rate was, again, in the same profile in the C horizon. The P index varied between 84.31and 71.29. It tended to decrease with depth for all profiles, except for the

Bss horizon of profile PII.

The bases/R2O3 value was below 1.4 in all profiles, ranging between 0.45 and 1.35. Generally, this index value increased with depth in all profiles except for the Bss1 horizon of profile PIV, including the low level of the index value.

Because of its mobility in the course of chemical weathering, alkaline and alkaline earth elements are able to undergo prominent changes, which resulted in the lower values on the Vogt index (V) for the unweathered samples. Some alkaline

and alkaline earth elements were flushed out of weathered grain material by rainfall.

### 3.5. Mass-Balance

The pedological mass-balance model is an approach in quantitative geochemistry used to calculate chemical weathering and

loss/gain in soil formation. Because of their abundance in the soils – 93% of total elemental structure occurs as oxides – and their importance in the soil formation process (Langley-Turnbaugh and Bockheim, 1998), the behavior of eight elements were assessed in this research (Si, Al, Ca, Mg, Na, K, Mn and Fe), and the values for these elements are indicated in Table 5. The researcher calculated the enrichment/depletion factors to get long-term weathering rates of soils in relation to the immobile element contents (Egli *et al*., 2008). The enrichment/depletion factor is a ratio of the chemical concentration of an

element in the soil to its concentration in the parent material. The mass transport function is described as the mass fraction of an element added or subtracted from the system during weathering relative to the mass of the element originally existing in the parent material. Strain, mass fractions added to, or subtracted from, each horizon and the loss or gain of elements during pedogenesis was calculated in accordance with Eqn. (1) to (3). The calculation of the gains or losses of elements require the values of immobile elements. The open-system mass transport functions ($\tau$) and strains ($M_{jflux}$) are listed against depth for

each soil and element (Table 5).

The mass-balance calculations that utilize immobile elements require two important assumptions related to elemental components (White, 1995; Egli *et al*., 2001a, b). The first assumption includes the determination of the composition of the parent material. For soils developed at sites on bedrock, the potential errors are confined to local heterogeneities in the bedrock composition. It is more difficult to estimate the initial composition when soils have developed on sedimentary





parent materials such as cover beds. Consequently, the gains and losses are calculated based on the lowermost and least weathered soil horizon (Waroszewski, 2016).

Zirconium and titanium have been used as immobile indices, which are stable in the soil environment (Brimhall and Dietrich 1987; Chadwick *et al.*, 1990; Merritts *et al.*, 1992; Harden 1988). Titanium was chosen as the immobile element for

calculating volume changes (strain).

Mass-balance calculations for basalt soils were conducted with Ti as the immobile elements, while Zr was used as the immobile element in the granite and rhyolite. The $SiO_2$ and Ti content did not correlate to the basalt soils ($R^2=0.04$), indicating that Ti concentrations were largely unaffected by eolian additions. Mineral phases in the basalt soils that could not be derived from the basalt parent material, such as quartz and mica, were classified as eolian materials. The feldspars in the

eolian material is likely to occur, and Al enrichment values in the soils support the addition of eolian feldspars (Heckman and Rasmussen, 2011). Mass-balance calculations of enrichment in Al were measured in the most horizons.

Element losses are seen in all upper horizons of the profiles except for profile PIII and PIV in Al and Na, and profile PII in Al, Mg, K and Fe. The most intense leaching was observed in the  Bss1 horizon (PIV) for Ca, Mg and Na. Negative mass-balance values of up to 89% were found for Ca in the illuvial horizon of this profile. However, an enrichment of Al (except

for PII) and Fe (except for PIV) were measured in the uppermost horizons.

Negative strains were measured in the PIV and PI profiles (except for Al in profile PIV and Fe in profile PI), while positive strains were measured in profiles PII and PIII (except for Ca in PIII, and Na in PIII and PII) (see Table 5). Positive values show dilatation due to the formation of humus or (bio) pores (Egli *et al.*, 2001b).

Greater losses of Ca and Mg emerge on the soil surface due to intense weathering and leaching, and losses increase with

depth in profiles PIand PIV. These elements appear to be the most mobile elements since they are the first elements to be leached from the weathering profile.

The majority of loss in the basalt bedrock mass was in the form of Ca ($-73.65 \, kg \, m^{-2}$), Mg ($-12.74 \, kg \, m^{-2}$), Na ($-1.18 \, kg \, m^{-2}$) and K ($1.79 \, kg \, m^{-2}$), sourced from the weathering of Ca feldspars, labrador and bytownite, that include high amounts of the element calcium (Welch and Ullman, 1993). There is a very small amount of sodium in basaltic rock. The reason for the

loss can be due to the small amount of this element. Ca and Mg have a significantly different behavior. The net losses of Ca and Mg seem in many cases to be significant when compared to Na. The open-system mass transport functions were slightly lower for base cations in all the profiles, and finally, the leaching of base cations was more noticeable.






Table 5. Mass transport function values (τ) and mass losses/gains (g.cm$^{-2}$) for some elements of soil profiles

| Profile -Land Position | Horizon | Si (τ) | Si $M_{iflux}$ | Al (τ) | Al $M_{iflux}$ | Ca (τ) | Ca $M_{iflux}$ | Mg (τ) | Mg $M_{iflux}$ | Na (τ) | Na $M_{iflux}$ | K (τ) | K $M_{iflux}$ | Fe (τ) | Fe $M_{iflux}$ |
|---|---|---|---|---|---|---|---|---|---|---|---|---|---|---|---|
| PIV Footslope | Ap | -0,03 | -4,62 | 0,19 | 6,34 | -0,76 | -22,09 | -0,47 | -4,03 | 0,04 | 0,14 | -0,01 | -0,06 | -0,05 | -1,06 |
| | Bss1 | -0,04 | -14,66 | 0,19 | 12,16 | -0,89 | 48,36 | -0,45 | -7,20 | 0,13 | -0,88 | 0,14 | -1,23 | -0,05 | -1,92 |
| | Bss2 | -0,03 | -7,39 | 0,26 | 13,62 | -0,07 | -3,20 | -0,12 | -1,51 | 0,08 | -0,44 | 0,07 | -0,51 | -0,02 | -0,70 |
| | C | 0,00 | 0,00 | 0,00 | 0,00 | 0,00 | 0,00 | 0,00 | 0,00 | 0,00 | 0,00 | 0,00 | 0,00 | 0,00 | 0,00 |
| | Total | - | -26,67 | - | 32,13 | - | -73,65 | - | -12,74 | - | -1,18 | - | -1,79 | - | -3,68 |
| PIII Backslope | Ap | 0,07 | 10,37 | 0,17 | 5,58 | -0,41 | -7,82 | 0,10 | 1,22 | -0,29 | -0,53 | 0,42 | 1,80 | 0,06 | 2,66 |
| | Cr | 0,00 | 0,00 | 0,00 | 0,00 | 0,00 | 0,00 | 0,00 | 0,00 | 0,00 | 0,00 | 0,00 | 0,00 | 0,00 | 0,00 |
| | Total | - | 10,37 | - | 5,58 | - | -7,82 | - | 1,22 | - | -0,53 | - | 1,80 | - | 2,66 |
| PII Low land plateau | A | -0,06 | -5,43 | 0,18 | 4,51 | -0,10 | -0,26 | 0,45 | 1,25 | 0,18 | -0,44 | 0,01 | 0,05 | 0,48 | 5,91 |
| | Bss1 | 0,03 | 5,20 | 0,39 | 21,44 | 0,13 | 0,72 | 3,04 | 17,98 | 0,12 | 0,66 | 0,27 | 2,86 | 0,82 | 21,86 |
| | Bss2 | 0,12 | 31,88 | 0,34 | 24,95 | 0,15 | 1,12 | 0,58 | 4,52 | -0,13 | -0,91 | -0,01 | -0,09 | 0,35 | 12,20 |
| | C | 0,00 | 0,00 | 0,00 | 0,00 | 0,00 | 0,00 | 0,00 | 0,00 | 0,00 | 0,00 | 0,00 | 0,00 | 0,00 | 0,00 |
| | Total | - | 31,65 | - | 50,90 | - | 1,59 | - | 23,74 | - | -0,69 | - | 2,81 | - | 39,97 |
| PI Shoulder | A | -0,08 | -9,22 | -0,13 | -2,41 | -0,20 | -2,98 | -0,20 | -2,45 | -0,15 | -0,35 | 0,05 | -0,38 | 0,10 | 2,39 |
| | Cr | 0,00 | 0,00 | 0,00 | 0,00 | 0,00 | 0,00 | 0,00 | 0,00 | 0,00 | 0,00 | 0,00 | 0,00 | 0,00 | 0,00 |
| | Total | - | -9,22 | - | -2,41 | - | -2,98 | - | -2,45 | - | -0,35 | - | -0,38 | - | 2,39 |

### 3.6. Clay minerals

5  In order to identify clay and primer minerals for Lithic Ustorthent and Typic Haplustert soils, XDR analysis and SEM (Scanning Electron Microscope) images of clay minerals and primer minerals of some soil samples were carried out and are shown in Figure 3 and Figure 4.

Basalt largely consists of mafic silicate minerals. With increasing weathering, these minerals were either transformed into a new mineral phase (e.g., smectite), or decomposed (Egli *et al.*, 2008). Random inspection of mineral composition detected 10  some non-clay minerals like quartz and K feldspar. The XRD patterns from the horizons of all profiles exhibited diffraction peaks of quartz (0.336 to 0.338) and from the Ap horizon of CD-P4 profile K feldspar (0.344) (Figure 3). While basaltic rocks may include quartz as an additional mineral, if considerable amounts are found in basaltic soils, they are usually of aeolian origin (Wilson, 2006).

The clay mineralogical components of the soils in all profiles were similar, but varied in abundance. The prepared samples 15  for XRD pattern analysis of K+25 $^{o}$C, K+550 $^{o}$C, Mg− air dried and Mg+ glycolated samples indicated kaolinite, smectite and illite in the C horizon of the PIV profile. In addition, vermiculite was also identified in the same layer.



Sm: Smectite, I: illite, K: kaolinite, Q: quarts

Figure 3. X-ray diffractograms of surface and subsurface horizons for each profiles




XRD patterns of some typical soil clay samples from the least and the strongest weathered soil horizons of sites are indicated in Figure 3. Kaolinite is present all samples; the most characteristic peaks of kaolinite: 0.72 nm and 0.36 (Moore and Reynolds, 1989), are not affected by glycolation, but disappear at $550\,^{\circ}$C. These peaks (001, 002), of the crystalline type, are sharp and symmetrical. The soils on the higher position had more kaolinite than those on the lower position, which is

5   indicative of the different developmental stages of the soils.

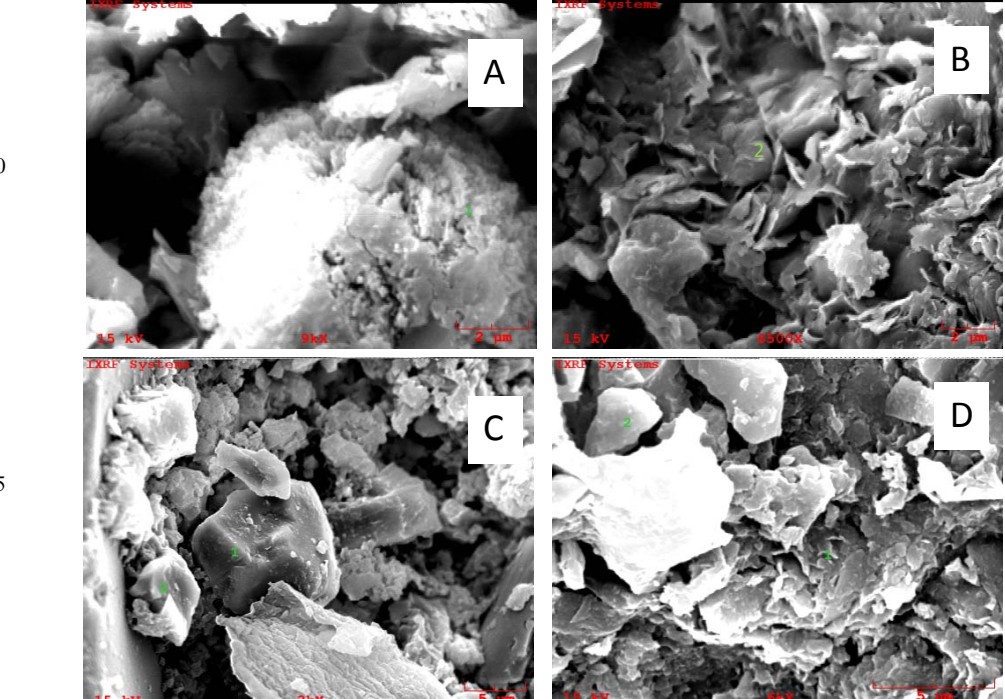

Figure 4: SEM images of clay minerals and primer mineral of some soil samples (A: 1: Quartz; 2: K-Feldspat;  B: 1: Illite-
20   Smectite; 2: Quartz; C: 1: Zeolite; 2: Smectite)

The K+25 $^{\circ}$C (air-dried) pattern exhibits a broad peak between 1.2 and 1.1, and 0.42 which includes smectite. These peaks can be separated into several peaks under Mg+glycolated conditions. The most important reflections (1.6 nm and 1.5 nm) correspond to smectites. Illite is distinguished by the (00l) series 1.01 and 0.50 nm. The air-dried patterns of these peaks, of

25   the crystalline type, are exhibited sharply and symmetrically. This clay mineral is unaffected by glycolation and heat treatment (550 $^{\circ}$C).

Kaolinite and smectite are the most frequently observed secondary phyllosilicates in magmatic soils, with kaolinite derived from the weathering of feldspar, and vermiculite from the weathering of mica (Wilson, 2006). Vermiculites with hydrous Al



are common during weathering of muscovite (Barnhisel and Bertsch, 1989). The present of hydrated kaolin phases might have resulted from the generally hot and dry climate conditions, and the ustic soil moisture regime. Both vermiculite and illite existed in small amounts in the subsurface horizons, however only illite was found in surface horizons. Illites, or clay micas, are generally inherited from the parent material or formed through the weathering of coarser mica particles (Allen and

Hajek, 1989). Researchers have also argued that illite may form authigenically (Harder, 1974; Norris and Pickering, 1983), or through the conversion of smectite to a 1 nm mineral in arid environments (Mahjoory, 1975).

## 4. Conclusions

In this study the features of pedogenic evolution of four soil profiles formed in topographically different positions of the southwestern–northeastern direction were investigated. The investigation considered the mass-balance, geochemical, clay mineralogy and chemical weathering indices of Typic Haplustert and Lithic Ustorthent soils represented by four profiles formed on basaltic parent material to understand the relationship between particular soils, and the landscapes and ecosystems in which they function. For this purpose, geochemical features, some physicochemical and mineralogical properties were

determined to compare the profile weathering rates. The results of the study showed a strong relationship between the topography and some of the soil's morphological, mineralogical, physical and chemical characteristics.

Soil depth and physical soil features such as texture, structure and bulk density were found to improve downwards within the toposequence. Fine earth concretions were lower in the back slope position, whereas, low slope degree positions included less gravel content.

Mass-balance calculations indicated that massive mineral weathering led to considerable Si losses through leaching and an exchange of cation, such as $Na^+$, $K^+$, $Ca^{2+}$ and $Mg^{2+}$, particularly from the upper horizons. These losses were particularly observed in the early stages of the soil's pedological development.

The soils were affected by element losses. However, they were still measurable. Mass-balance calculations and the use of weathering indices showed significant leaching of major base cations and Si owing to the weathering of primary minerals

such as feldspars and olivine. The prevailing processes which were determined by the mass-balance analysis included desilication and loss of bases. In general, negative values and element loss increased due to progressing soil age.

Ultimately, chemical weathering of silicate rocks through hydrolysis leads to an exchange of the cations $Na^+$, $K^+$, $Ca^{2+}$ and $Mg^{2+}$ for $H^+$ and perhaps a loss of $Si^{4+}$ (Kramer, 1968). Cations such as $Na^+$, $K^+$ and $Ca^{2+}$ are commonly generated by the weathering of feldspar. The cation $Mg^{2+}$ is derived from sheet silicates and mafic minerals, and it resides in smectite clays

(Nesbitt and Young, 1984; Pettijohn *et al.*, 1987). In general, hydrolytic weathering causes a progressive transformation of affected components into clay minerals, ultimately kaolinite.

The weathering indices values from CIA, CIW, WIP, PIA, P bases/$R_2O_3$ and V indicated that soils classified as Entisol and Vertisol have similar pedochemical properties. In spite of similar weathering rates, the soils were classified differently as a





result of erosion. This showed that the conditions of soil development in the study area had a far more impact on elemental loss and weathering than the parent material in the site.

## 4. Acknowledgments

The authors gratefully acknowledge the scientific research grant (TÜBİTAK-213O073) from the Scientific and Technological Research Council of Turkey.

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
