# Peer review of "Geochemical mass-balance, weathering and evolution of soils formed on a Quaternaryage basaltic toposequences"

_Solid Earth, 2016_

## Referee Comment (RC3)

Solid Earth Manuscript Number: se-2016-105

Geochemical mass-balance, weathering and evolution of soils formed on a Quaternary age basaltic toposequences

by Hüseyin ŞENOL et al.

General:

The manuscript refers to the weathering intensity, element release and soil development of four profiles in the Black sea Region of Turkey.

The paper is well written and quite well organized and provides valuable information for the understanding of rock weathering and soil formation. Some sections of the text that are in the results and discussion should be placed in the introduction or the methods rather than in the results. For example, the paragraphs 3.4, page 11, lines 10-33 and 3.5, page 13, lines 19-35 present the different weathering indices and the mass balance calculation.

The authors have used several weathering indices to quantify weathering intensity and elemental loss. The CIA is mainly used to estimate weathering intensity of felsic rocks with important release of alkaline and alkaline-earth cations, release of silica and relative accumulation of aluminum from the weathering of feldspars, as written by the authors. This index has some limits as it does not take into account the loss in silica and the accumulation of iron in the soils. The authors should also refer to a recent article - The weathering intensity scale (WIS): an alternative approach of the chemical index of alteration (CIA), Meunier et al., 2013 - that uses both loss in silica and accumulation of iron and aluminum to quantify weathering intensity. This approach allows a better discrimination of highly weathered rocks than CIA and related indices.

To use this approach to quantify weathering the authors should pay attention to two important factors: 1) the reference parent rock should be unweathered (The CIA values of the basalt are in the range of what observed for unweathered basalt), 2) there should be a genetic relationship between the soil, the saprolite and the parent material; i.e. an homogeneity of the profile without allochthonous deposit on the surface.

To summarize, this manuscript present data that are suitable for publication in Solid Earth. However, I guess the manuscript is not acceptable in its present from; several issues need to be addressed. Therefore, I strongly recommend the authors to perform minor revision of their manuscript to be in order for publication in SE.

Specific comments

Page 4, line 1: add a.s.l "above sea level" or a.m.s.l "above mean sea level" after the values of the elevation even if it is implicit.

Page 4, line 7: add $yr^{-1}$ after the values of the mean annual rainfall.

Page 4, lines 15-16: The transverse illustrated in Figure 2 should be also shown in Figure 1 if possible.

Page 5, line 6: replace NaOAC by sodium acetate or give the formulae.

Page 5, line 12: replace $NH_4OAC$ by ammonium acetate or give the formulae.

Page 5, lines 14-16: give more details of the extraction of the clay fraction after the dispersion method. The procedure of clay fraction treatments should be given before the XRD as they are done before recording XRD patterns.

Page 8, line 33: the unit of CEC is $cmol(c).kg^{-1}$ (add comma)

Page 12, table 4: Add the caption of the table and replace Verisol by Vertisol for profile PIV / footslope

Page 15, line 5: write In order to identify primary minerals and clay minerals

Page15, lines 10-11: check the position of the peaks of quartz. Quartz has a peak at 3.34 Å but not at 3.36 and 3.38 Å. The position of the peaks for all minerals should be carefully checked.

Page 16, figure 3: The figure should be clarified and simplified. The correct spelling is quartz in the figure caption. The patterns should be in the order Mg, Mg-Gly, K25 and K550 from the bottom to the top of the plot to show more easily changes in peak position with the different treatments. Only the main peaks should be labeled for better clarity. Some values are given for features that are rather "noise" of the pattern rather than peaks. e.g. cr: for the 2:1 swelling minerals the peaks at 1.85 Å EG, 1.3 Å Mg, 1.26 Å Mg, 1.12 Å K550 should not be given in the figure. In the text it is possible to explain with asymmetry of shoulder of the main peaks; PII: delete the peaks at 1.91 Å EG, 1.48 Å Mg, 1.38 Mg, 1.16 Å K550. Same remarks for all plots.

Check the labeling of the peaks. For PI the peak of the 2:1 swelling clay mineral labeled at 16.8 Å is not at this d-spacing. 16.8 Å with CuK$\alpha$ radiation is at about 5.25°2$\theta$ which is obviously not the case in this figure, but rather at 5.8 °2$\theta$ namely ~15 Å. Compare with other samples.

The values of the peaks of quartz and kaolinite should be given with higher precision. The 002 of kaolinite is at 3.58 Å and the peak of quartz at 3.34 Å.

Reference cited

Meunier, A., Caner, L., Hubert, F., El Albani, A., and Prêt, D. (2013). The weathering intensity scale (WIS): an alternative approach of the chemical index of alteration (CIA). American Journal of Science, 313, 113–143.

The article can be provided upon request as well as the calculation procedure.

---

## Referee Comment (RC1) · O. Başkan (Referee) · 26 Sep 2016

Dear Editor, This paper (doi:10.5194/se-2016-105) written by Hüseyin ŞENOL, Tülay TUNÇAY and Orhan DENGİZ deals with important research about "Geochemical mass-balance, weathering and evolution of soils formed on a Quaternaryage basaltic toposequences". This study was carried out in the central Black Sea region of Turkey and aimed to assess the geochemical mass-balance and weathering intensity of Typic Haplustert and Lithic Ustorthent soils that developed in a Quaternary-age basaltic toposequence under semi-humid conditions. As well known that soil has complex system and explanation and determination of 'soil production' or 'soil formation' are difficult. There are several approaches and concepts exist that lead to potentially dif-

ferent or possibly even contradictory results. In this study, an empirical method was used by determining the weathering rates and mass-balance of soils developed on a catena with respect to the soil-forming factor topography, and using some weathering indices based on geochemical information. Therefore this study has assessed the role of understanding about soil formation over a toposequence and Quaternary-age basaltic parent materials. Moreover, this case is still able to add some substantial knowledge of interest to not only soil scientists but also to earth scientists and environmental readerships. In addition, the paper contains some useful information that is worthy of publication and usefulness for other researchers in this field. Statement of manuscript is clarity. The main objectives of this research were given by authors in introduction section and these objectives were clarity explained in result and discussion section. Methodology is suitable for research aims and analytical results of research were given adequately in result and discussions section. On the other hand, some additional information should be given in test, - It should be given all math formulas as high quality - Abbreviation under Table 2 , O.M: organic Matter should be O.M: Organic Matter - In all tables should be the same standard using dot "." or comma "," - It should be checked all references in text with reference list. There are some literatures in text such as Dengiz et al., 2013; Dengiz, 2007; Sommer et al., 2008 etc. on the other hand, these literatures should be added in reference section I think that these suggestions contribute to the manuscript. In brief, my opinion is, this manuscript has been written in standard scientific way and is suitable for publication in an international journals as like Journal of Soil Earth. Consequently, the paper has some potential, but needs quite some work. Therefore, the manuscript can be accepted for publication after minor revision.

Sincerely yours,
* * *

---

## Referee Comment (RC2) · Anonymous Referee #2 · 5 Oct 2016

Dear Editor,

In this paper, characteristics of 4 soil profiles on Quaternary-age basaltic toposequence of Middle Black Sea region of Turkey. It can provide contributions to scientific literature and data were well-organized. This manucript may be needs minor revisions. In present form of this paper after minor revision to accept publication in your journal as a research paper.

**1)** A reference about meteorological data should be provided in Section 2.3.

**2)** Wording errors in Table 3 should be corrected. (Correct form is $SiO_2$, $Al_2O_3$, $Fe_2O_3$, $Na_2O$, $K_2O$, $TiO_2$, $P_2O_5$, $Cr_2O_3$).

**3)** What does the image D of Figure 4 represent?

**4)** The date for Moore and Reynolds in section 3.6 and the data provided in references are different. Correct the date in references.

**5)** The following literatures were listed in reference section even they were referred in paper. Include them in reference section (Middleburg et al., 1988; Dengiz et al., 2013; Bohn et al., 1985; Sommer et al., 2008; Jong et al., 2000; Dengiz, 2007; Brimhall et al., 1988; 1991, Yair, 1990; Dahlgren et al., 1997; Canton, 2003; Dengiz, 2010; Kreznor et al., (1989); Rezaei and Gilkes (2005); Dengiz et al., (2013); FAO/ISRIC, 2006; Aksoy (1991); Jianwu et al., (2014); Souri et al., 2016; Welch and Ullman, 1993; Moore and Reynolds, 1989; Pettijohn et al., 1987).

**6)** The reference "Buol and Hole, 1961" was not referred in the text.

**7)** The reference "Egli et al. (2008)" was written two times. Delete one of them.

Sincerely yours,

---

## Short Comment (SC1) · 14 Oct 2016

Dear authors:

I just would like to make some recommendations in order to improve the manuscript. Of course, these are kind suggestions to you.

(1) I strongly recommend shortening the title. The method is the key of your research, so I suggest this: "Geochemical mass-balance applied to the study of weathering and evolution of soils".

(2) At the beginning of the abstract, it is a good idea to include a pair of sentences to highlight the relevance of studying pedogenesis and soils as an open system. Then,

the purpose.

(3) The final paragraph of the introduction must include the objectives, as you have done. But I suggest stating what is the main objective (suggesting an emprirical method for the study of ... involving mass-balance study) and secondary objectives as [i], [ii], [iii], [iv], ...

(4) For figures, I strongly encourage you to use color because [i] it improves readibility and [ii] it attracts citations.

I have submitted a separate file (find it as a supplementary file) with some annotations for the first sections of your manuscript. I hope you find it useful.

Antonio Jordán

Executive Editor of Solid Earth

Please also note the supplement to this comment:
http://www.solid-earth-discuss.net/se-2016-105/se-2016-105-SC1-supplement.pdf

---

## Author Comment (AC1) · 14 Oct 2016

Dear Editor; According to Referee's correction; 1- The abbreviations of Table 2 were given as OM: Organic Matter. The explanations under Table 2 Ariel 8 letter changed to Times New Roman 8. In Table 3, the commas turned to dot. In Table 5, land position column is deleted due to the replication. In Table 5 the commas turned to dots.

2- Some literatures were added in references section.

All the corrections are shown with red.

Please also note the supplement to this comment:

[Figure]

http://www.solid-earth-discuss.net/se-2016-105/se-2016-105-AC1-supplement.pdf

---

## Author Comment (AC2) · 14 Oct 2016

Dear Editor; According to minor corrections that referees recommend me have been corrected. The corrections are referee given below: 1- Meteorological data's reference is provided in Section 2.3. 2- Wording errors in Table 3 is corrected as you suggested. The commas turned to dot. % symbol used in one lin. In Table 3 Depth and Bulk Density columns are deleted due to given in Table 2. 3- The image D was written as D: 2: Smectite in Figure 4. 4- The reference of Moore and Reynolds dates is corrected as 1997. 5- The literatures are listed in reference section. 6- The reference "Buol and Hole, 1961" is deleted from the reference section. 7- One of reference "Egli et al., 2008" is deleted. All the corrections are shown with red.

Please also note the supplement to this comment:
http://www.solid-earth-discuss.net/se-2016-105/se-2016-105-AC2-supplement.pdf
* * *
[Figure]

**Supplement:**

[revised manuscript text omitted]

FAO/ISRIC. World references base for soil resources. World Soil Report No, 103. Rome, 2006.

Fedo, C.M., Nesbitt, H.W. and Young, G.M.: Unraveling the effects of potassium metasomatism in sedimentary rocks and paleosols, with implications for paleoweathering conditions and provenance. Geology, 23: 921-924, 1995.

Fiantis, D., Nelson, M., Shamshuddin, J., Goh, T.B. and Van Ranst, E.: Determination of the Geochemical Weathering Indices and Trace Elements Content of New Volcanic Ash Deposits from Mt. Talang (West Sumatra) Indonesia. Eurasian Soil Science, 43, 13, 1477-1485, doi:10.1134/S1064229310130077, 2010.

Gee, G.W. and Bauder, J.W.: Particle-size Analysis. P. 383 - 411. In A.L. Page (ed.). Methods of soil analysis, Part 1, Physical and mineralogical methods (Second Edition), Agronomy, 383-411, doi: 10.12691/env-3-4-4, 1986.

Göl, C. and Dengiz, O.: Effect of Modifying Land Cover and Long-Term Agricultural Practices on the Soil Characteristics in Native Forest-Land. Journal of Environmental Biology, 29,5, 667-682, PMID:19295064, 2008.

Gupta, R.D., Arora, S., Gupta, G.D. and Sumberia, N.M.: Soil Physical Variability in Relation to Soil Erodibility under Different Land Uses in Foothills of Siwaliks in N-W India. Tropical Ecology, 51, 2, 183-197, 2010.

Harden, J.W.: Genetic interpretations of elemental and chemical differences in a soil chronosequence, California. Geoderma, 43, 179-193, doi:10.1016/0016-7061(88)90042-0, 1988.

Harder, H.: Illite mineral synthesis at surface temperatures. Chemical Geology, 14, 241–253, doi: 10.1016/0009-2541(74)90062-X, 1974.

Harnois, L.: The CIW index- a new chemical index of weathering. Sedimentary Geology, 55, 319-322, doi: 10.1016/0037-0738(88)90137-6, 1988.

Heckman, K. and Rasmussen, C.: Lithologic controls on regolith weathering and mass flux in forested ecosystems of the southwestern USA. Geoderma, 164, 99-111, doi: 10.1016/j.geoderma.2011.05.003, 2011.

Jianwu, L., Ganlin, Z. and Zitong, G. Mobilization and redistribution of elements in soils developed from extreme weathering basalt on Hainan Island. Chinese Journal of Geochemistry, 33, 262-271, DOI: 10.1007/s11631-014-0686-y, 2014.

Jong, E., Pennock, D.J. and Nestor, P.A.: Magnetic susceptibility of soils in different slope positions in Saskatchewan, Canada. Catena, 40, 291-305, DOI: 10.1016/S0341-8162(00)00080-1, 2000.

Kramer, J.R.: Mineral-water equilibria in silicate weathering. XXIII. International Geological Congress, 6,149-160, 1968.

Kreznor, W.R., Olson, K.R., Banwart, W.L. and Johnson, D.L.: Soil, landscape, and erosion relationships in a northwest Illinois watershed. Soil Science Society Americal Journal, 53, 1763-1771, doi:10.2136/sssaj1989.03615995005300060026x, 1989.

[revised manuscript text omitted]

---

## Author Comment (AC3) · 18 Oct 2016

Dear Editor; Thank you for recommendations in order to improve the manuscript. The recommendations are corrected as below: 1- The title is shortened as "Geochemical mass-balance applied to the study of weathering and evolution of soils" 2- Before the purpose a sentence is added to highlight the relevance of the study pedogenesis and soils as an open system. 3- Objectives were given as main and second in the final paragraph of the introduction. 4- In Figure 1, the map is colored and corrected as recommended. Such as space between Mediterranean Sea, Black Sea and boards of the Turkey's map

All the corrections are shown with red.

[Figure]

Please also note the supplement to this comment:
http://www.solid-earth-discuss.net/se-2016-105/se-2016-105-AC3-supplement.pdf
* * *
Interactive
comment

[Figure]

**Supplement:**

**Geochemical mass-balance applied to the study of weathering and evolution of soils**

Hüseyin ŞENOL[1*], Tülay TUNÇAY[2], Orhan DENGİZ[3]

[1]Suleyman Demirel University, Faculty of Agriculture, Department of Soil Science and Plant Nutrition, Isparta, Turkey.
[2]Soil Fertilizer and Water Resources Central Research Institute, Ankara, Turkey.
[3]Ondokuz Mayıs University, Faculty of Agriculture, Department of Soil Science and Plant Nutrition, Samsun, Turkey.

*Correspondence to*: Huseyin Senol (*huseyinsenol@sdu.edu.tr*)

**Abstract.** Soil is viewed as an open system with additions, losses, translocations and transformations of materials. The purpose of this research is to assess the geochemical mass-balance and weathering intensity of Vertisols (Typic Haplusterts) and Entisols (Lithic Ustorthents) developed in a Quaternary-age basaltic toposequence under semi-humid conditions in the central Black Sea region of Turkey. We used mass-balance analysis with a view to measuring elemental gains and losses along with alterations concerning soil forming processes. For this end, geochemical properties, elemental mass-balance changes and certain physicochemical features were identified to benchmark the weathering levels of the profiles. Lithic Ustorthents are distinguished by having a rough texture along with a low organic substance ingredient, whereas Typic Haplusterts have a high clay texture with low bulk density and slickenside features. X-ray diffraction showed that smectites were the prevailing minerals inside the Typic Haplusterts, while a significant amount of kaolinite and illite was observed in the Lithic Ustorthents. Mass-balance computations indicated that massive mineral weathering resulted in substantial Si losses through leaching as well as an exchange of cations, such as $Na^+$, $K^+$, $Ca^{2+}$ and $Mg^{2+}$, particularly from the upper horizons. The study also took into account other features such as the pedogenic evolution of soils using weathering indices such as CIA (chemical index of alteration), CIW (Chemical Index of Weathering), bases/$R_2O_3$, WIP (Weathering Index of Parker), P (Product index), PIA (Plagioclase index of alteration). A
[revised manuscript text omitted]

FAO/ISRIC. World references base for soil resources. World Soil Report No, 103. Rome, 2006.

Fedo, C.M., Nesbitt, H.W. and Young, G.M.: Unraveling the effects of potassium metasomatism in sedimentary rocks and paleosols, with implications for paleoweathering conditions and provenance. Geology, 23: 921-924, 1995.

Fiantis, D., Nelson, M., Shamshuddin, J., Goh, T.B. and Van Ranst, E.: Determination of the Geochemical Weathering Indices and Trace Elements Content of New Volcanic Ash Deposits from Mt. Talang (West Sumatra) Indonesia. Eurasian Soil Science, 43, 13, 1477-1485, doi:10.1134/S1064229310130077, 2010.

Gee, G.W. and Bauder, J.W.: Particle-size Analysis. P. 383 - 411. In A.L. Page (ed.). Methods of soil analysis, Part 1, Physical and mineralogical methods (Second Edition), Agronomy, 383-411, doi: 10.12691/env-3-4-4, 1986.

Göl, C. and Dengiz, O.: Effect of Modifying Land Cover and Long-Term Agricultural Practices on the Soil Characteristics in Native Forest-Land. Journal of Environmental Biology, 29,5, 667-682, PMID:19295064, 2008.

Gupta, R.D., Arora, S., Gupta, G.D. and Sumberia, N.M.: Soil Physical Variability in Relation to Soil Erodibility under Different Land Uses in Foothills of Siwaliks in N-W India. Tropical Ecology, 51, 2, 183-197, 2010.

Harden, J.W.: Genetic interpretations of elemental and chemical differences in a soil chronosequence, California. Geoderma, 43, 179-193, doi:10.1016/0016-7061(88)90042-0, 1988.

Harder, H.: Illite mineral synthesis at surface temperatures. Chemical Geology, 14, 241–253, doi: 10.1016/0009-2541(74)90062-X, 1974.

Harnois, L.: The CIW index- a new chemical index of weathering. Sedimentary Geology, 55, 319-322, doi: 10.1016/0037-0738(88)90137-6, 1988.

Heckman, K. and Rasmussen, C.: Lithologic controls on regolith weathering and mass flux in forested ecosystems of the southwestern USA. Geoderma, 164, 99-111, doi: 10.1016/j.geoderma.2011.05.003, 2011.

Jianwu, L., Ganlin, Z. and Zitong, G. Mobilization and redistribution of elements in soils developed from extreme weathering basalt on Hainan Island. Chinese Journal of Geochemistry, 33, 262-271, DOI: 10.1007/s11631-014-0686-y, 2014.

Jong, E., Pennock, D.J. and Nestor, P.A.: Magnetic susceptibility of soils in different slope positions in Saskatchewan, Canada. Catena, 40, 291-305, DOI: 10.1016/S0341-8162(00)00080-1, 2000.

Kramer, J.R.: Mineral-water equilibria in silicate weathering. XXIII. International Geological Congress, 6,149-160, 1968.

Kreznor, W.R., Olson, K.R., Banwart, W.L. and Johnson, D.L.: Soil, landscape, and erosion relationships in a northwest Illinois watershed. Soil Science Society Americal Journal, 53, 1763-1771, doi:10.2136/sssaj1989.03615995005300060026x, 1989.

[revised manuscript text omitted]

---

## Author Comment (AC4) · 25 Oct 2016

Dear Editor, Thank you for recommendations in order to improve the manuscript. The recommaditions are corrected as below;

Page 4, line 1: To the Line 1 "a.m.s.l" is added. Page 4, line 7: To the Line 7 "yr-1" is added. Page 4, lines 15‐16: Figure 2 is suitable as present Page 5, line 6: NaOAc is replaced to sodium acetate. Page 5, line 12: NH4OAC is replaced to ammonium acetate Page 5, lines 14‐16: Present details are sufficient. Page 8, line 33: For the CEC unit the comma is added. Page 12, table 4: Verisol is replaced to Vertisol (Table3 and 4) Page 15, line 5: In the second paragraph primary and clay minerals are identified. Page15, lines 10‐11: Peaks are checked and the patterns are in orde

[Figure]

Mg, Mg‐Gly, K250 and K25 from bottom to the top.

The reference of Meunier et all. (2013) is added to the CIA indices.

Please also note the supplement to this comment:
http://www.solid-earth-discuss.net/se-2016-105/se-2016-105-AC4-supplement.pdf

**Supplement:**

**Geochemical mass-balance applied to the study of weathering and evolution of soils**

Hüseyin ŞENOL[1*], Tülay TUNÇAY[2], Orhan DENGİZ[3]

[revised manuscript text omitted]

FAO/ISRIC. World references base for soil resources. World Soil Report No, 103. Rome, 2006.

Fedo, C.M., Nesbitt, H.W. and Young, G.M.: Unraveling the effects of potassium metasomatism in sedimentary rocks and paleosols, with implications for paleoweathering conditions and provenance. Geology, 23: 921-924, 1995.

30 Fiantis, D., Nelson, M., Shamshuddin, J., Goh, T.B. and Van Ranst, E.: Determination of the Geochemical Weathering Indices and Trace Elements Content of New Volcanic Ash Deposits from Mt. Talang (West Sumatra) Indonesia. Eurasian Soil Science, 43, 13, 1477-1485, doi:10.1134/S1064229310130077, 2010.

Gee, G.W. and Bauder, J.W.: Particle-size Analysis. P. 383 - 411. In A.L. Page (ed.). Methods of soil analysis, Part 1, Physical and mineralogical methods (Second Edition), Agronomy, 383-411, doi: 10.12691/env-3-4-4, 1986.

Göl, C. and Dengiz, O.: Effect of Modifying Land Cover and Long-Term Agricultural Practices on the Soil Characteristics in Native Forest-Land. Journal of Environmental Biology, 29,5, 667-682, PMID:19295064, 2008.

Gupta, R.D., Arora, S., Gupta, G.D. and Sumberia, N.M.: Soil Physical Variability in Relation to Soil Erodibility under Different Land Uses in Foothills of Siwaliks in N-W India. Tropical Ecology, 51, 2, 183-197, 2010.

Harden, J.W.: Genetic interpretations of elemental and chemical differences in a soil chronosequence, California. Geoderma, 43, 179-193, doi:10.1016/0016-7061(88)90042-0, 1988.

Harder, H.: Illite mineral synthesis at surface temperatures. Chemical Geology, 14, 241–253, doi: 10.1016/0009-2541(74)90062-X, 1974.

Harnois, L.: The CIW index- a new chemical index of weathering. Sedimentary Geology, 55, 319-322, doi: 10.1016/0037-0738(88)90137-6, 1988.

Heckman, K. and Rasmussen, C.: Lithologic controls on regolith weathering and mass flux in forested ecosystems of the southwestern USA. Geoderma, 164, 99-111, doi: 10.1016/j.geoderma.2011.05.003, 2011.

Jianwu, L., Ganlin, Z. and Zitong, G. Mobilization and redistribution of elements in soils developed from extreme weathering basalt on Hainan Island. Chinese Journal of Geochemistry, 33, 262-271, DOI: 10.1007/s11631-014-0686-y, 2014.

Jong, E., Pennock, D.J. and Nestor, P.A.: Magnetic susceptibility of soils in different slope positions in Saskatchewan, Canada. Catena, 40, 291-305, DOI: 10.1016/S0341-8162(00)00080-1, 2000.

Kramer, J.R.: Mineral-water equilibria in silicate weathering. XXIII. International Geological Congress, 6,149-160, 1968.

Kreznor, W.R., Olson, K.R., Banwart, W.L. and Johnson, D.L.: Soil, landscape, and erosion relationships in a northwest Illinois watershed. Soil Science Society Americal Journal, 53, 1763-1771, doi:10.2136/sssaj1989.03615995005300060026x, 1989.

Langley-Turnbaugh, S.J. and Bockheim, J.G.: Mass balance of soil evolution on late Quaternary marine terraces in coastal Oregon. Geoderma, 84, 265-364, 1998.

Machado, A., Azevedo, J.M.M., Almeida, D.P.M. and Chemale, JrF.: Geochemistry of volcanic Rocks from Faial island (Azores). Revista Electrónica de Ciências da Terra Geosciences On-line Journal, 5, 1-14, 2008.

Mahjoory, R.A.: Clay mineralogy, physical and chemical properties of some soils in arid regions of Iran. Soil Science Society of America Journal, 39, 1157-1164, doi: 10.2136/sssaj1975.03615995003900060036x, 1975.

Merritts, D.J., Chadwick, O.A., Hendricks, D.M., Brimhall, G.H. and Lewis, C.J.: The mass balance of soil evolution on late Quaternary marine terraces, northern California. Geological Society of America Bulletin, 104, 1456-1470, doi:10.1130/0016-7606(1992)1042.3.CO;2, 1992.

Meunier, A., Caner, L., Hubert, F., El Albani, A., and Prêt, D.: The weathering intensity scale (WIS): an alternative approach of the chemical index of alteration (CIA). American Journal of Science, 313, 113-143, doi: 10.2475/02.2013.03, 2013.

Middleburg, J.J., van der Weijden, C.H. and Woittiez, J. R.W.: Chemical processes affecting the mobility of major, minor and trace elements during weathering of granitic rocks. Chemical Geology, 68, 253-273, DOI: 10.1016/0009-2541(88)90025-3,1988.

[revised manuscript text omitted]